# Bioactivity Profiling of *Daedaleopsis confragosa* (Bolton) J. Schröt. 1888: Implications for Its Possible Application in Enhancing Women’s Reproductive Health

**DOI:** 10.3390/ph17050600

**Published:** 2024-05-08

**Authors:** Djordje Ilić, Maja Karaman, Mirjana Bogavac, Jovana Mišković, Milena Rašeta

**Affiliations:** 1Clinical Centre of Vojvodina, Department of Obstetrics and Gynecology, Faculty of Medicine, University of Novi Sad, Hajduk Veljkova 3, 21000 Novi Sad, Serbia; djordje.ilic@mf.uns.ac.rs (D.I.); mirjana.bogavac@mf.uns.ac.rs (M.B.); 2ProFungi Laboratory, Department of Biology and Ecology, Faculty of Sciences University of Novi Sad, Trg Dositeja Obradovića 2, 21000 Novi Sad, Serbia; jovana.maric@dbe.uns.ac.rs (J.M.); milena.raseta@dh.uns.ac.rs (M.R.); 3Department of Chemistry, Biochemistry and Environmental Protection, Faculty of Sciences, University of Novi Sad, Trg Dositeja Obradovića 3, 21000 Novi Sad, Serbia

**Keywords:** fungi, hemolytic, DNA fragmentation, enzyme inhibition, *Daedaleopsis confragosa*

## Abstract

This study investigates the bioactivity profile of wood-rotting fungal species *Daedaleopsis confragosa* (Bolton) J. Schröt. 1888, focusing on its antioxidant, cytotoxic, and genotoxic activities and enzyme modulation properties with respect to its possible application in terms of enhancing women’s reproductive health. Two types of extracts, including those based on EtOH extraction (DC) and hydrodistillation (DCHD), were investigated. The results indicate that the radical scavenging capacity against the DPPH radical and reduction potential were stronger in the DC extracts owing to the higher total phenolic content (TPC) and total flavonoid content (TFC) (25.30 ± 1.05 mg GAE/g d.w. and 2.84 ± 0.85 mg QE/g d.w., respectively). The same trend was observed in the protein phosphatase-1 (PP1) activity and in the genotoxic activity against the δ virus since only the DC extract exhibited DNA disintegration regarding a dilution of 1:100. Conversely, the DCHD extract exhibited increased hemolytic and cytotoxic effects (339.39% and IC_50_ = 27.76 ± 0.89 μg/mL—72 h incubation, respectively), along with greater inhibition of the AChE enzyme (IC_50_ = 3.11 ± 0.45 mg/mL) and hemolytic activity. These results suggest that terpenoids and steroids may be responsible for the observed activity in DCHD as these compounds could potentially be extracted following the HD procedure. This comprehensive bioactivity profiling offers valuable insights into the potential therapeutic applications of *D. confragosa* from Serbia and underscores the importance of further investigations for harnessing its pharmacological potential.

## 1. Introduction

In recent years, there has been a notable surge in the interest surrounding natural products due to their potential therapeutic applications, mostly in terms of the escalation of the challenges posed by the growing resistance of microorganisms to antibiotics, alongside the complexities associated with treating the rising incidences of cancerous, neurodegenerative, and various other diseases [1]. Fungi, in particular, have attracted considerable interest due to their ability to produce a myriad of bioactive compounds with diverse pharmacological properties, with a special emphasis on lignicolous macrofungi, mostly belonging to the *Polyporaceae* family [2,3,4,5], with diverse ecological and morphological features, known as decomposers of wood polymers, such as lignin, cellulose, and hemicelluloses, through the production of cell wall-degrading enzymes and non-enzyme compounds [5,6]. Among the ≈1.5 million fungal species, 10,000 belong to lignicolous macrofungi, including ≈5000 edible and over 1800 medicinal species [7]. Naturally found on plant debris and decaying wood materials, lignicolous fungi have been detected as one of the richest in terms of the content and diversity of secondary metabolites [5], including polyphenols, such as flavonoids, known for their antioxidant and antimicrobial properties [8]. Other mycochemicals exhibit beneficial antimicrobial, anticancerous, and anti-inflammatory activities, thus showing general pharmaceutical properties with minimal toxicity [7].

Acetylcholinesterase (AChE) and protein phosphatase-1 (PP1) are widely used in toxicological assessments across various species. AChE operates in both the peripheral and central nervous systems, while PP1 is crucial in regulating various biological processes, including carbohydrate and lipid metabolism [4,9]. Dysfunctions in their activity can lead to disease onset [10]. AChE inhibitors are used in treating Alzheimer’s and myasthenia gravis, while PP1 inhibitors aid Parkinson’s treatment, and activators help to manage diabetes mellitus and cancer [9]. Insulin activates PP1 in insulin-sensitive cells, regulating the glycogen synthase phosphorylation and glucose deposition in skeletal muscles [9]. Currently, there is a lack of literature data regarding inhibitors targeting PP1 derived from lignicolous macrofungi, while a great deal of data is available on AChE inhibitors [4,11].

In the context of gestational diabetes mellitus (GDM), fungi have shown potential in managing diabetes by supporting better glucose control and potentially reducing the risk of related complications. Mycelia powder extracted from *Agaricus brasiliensis* and *Ganoderma lucidum* could help to manage GDM by improving the insulin sensitivity and reducing the postprandial glucose levels [12]. The fungal polysaccharides, proteins, dietary fibers, lectins, lactones, alkaloids, terpenoids, sterols, and phenolic compounds have been shown to possess immune-modulatory and anti-mutagenic activities, which can be beneficial in managing GDM [12]. Furthermore, a randomized clinical trial found that a fungal-based diet could reduce the risk of pregnancy-induced hypertension and GDM, as well as control newborn birth weight and reduce comorbidities such as gestational weight gain and excessive gestational weight gain [13].

Additionally, many lignicolous fungal species like *G. lucidum*, *Grifola frondosa*, *Lentinula edodes*, and *Trametes versicolor* possess anticancer properties since they contain high-molecular-weight polysaccharides, mostly β-glucans, which have been proven to stimulate innate immune cells, such as monocytes, natural killers (NK) cells, and dendritic cells, having direct antitumor effects [14]. Among these, many other species, such as *Agaricus bisporus*, *Antrodia cinnamomea*, *Cordyceps sinensis*, *Cordyceps militaris*, and *Pleorotus ostreatus*, demonstrated anticancer properties against breast cancer [14], specifically in the estrogen-dependent cell lines, via a direct antiproliferative effect and reduction in the generation of reactive oxygen species [15,16,17]. In particular, the fungal extracts improved the chemotherapy efficacy in cancer cells by sensitizing the cancer cells to chemotherapy drugs, inducing apoptosis and antiproliferation [14]. However, fungi are well-known to act as biological response modifiers by bolstering the immune system, so more attention should be focused on their role in enhancing women’s health, specifically regarding the anticancer effects on breast or cervical tumors [14]. A case-control study on the consumption of dietary fungi and breast cancer risk showed that fungi may decrease the breast cancer risk in postmenopausal women [18], while previous data suggest that it can be related to lower risk for breast cancer among premenopausal women as well; this relation may be more robust among women with hormone receptor-positive tumors [19]. Fungal compounds such as lectins, ergosterol, and triterpenoids like ganodermanontriol and ganoderic acid, along with their derivatives, are pivotal in cancer therapy [20]. Lectins, in particular, have gained attention for their therapeutic potential, supported by both animal and human clinical trials [21]. Their mechanisms of action include initiating cytotoxicity and apoptosis, inducing cell cycle arrest, downregulating telomerase activity, blocking angiogenesis, and inhibiting tumor growth, often through specific binding to cancer cell membranes [20]. Ergosterol peroxide and its sulfonamide derivative seem to have a multifaceted impact on inflammatory breast cancer cells. The arrest of the cell cycle at the G1 phase, activation of caspase-3/7 leading to apoptosis, and attenuation of key proteins like AKT1, AKT2, BCL-XL, cyclin D1, and c-Myc all contribute to their anticancer properties [22]. Natural compounds like ergosterol peroxide and its derivatives offer the advantage of potentially lower side effects compared to synthetic drugs, which is especially significant for vulnerable populations like menopausal or pregnant women.

*Daedaleopsis confragosa* (Bolton) J. Schröt. 1888, a wood-rotting fungus commonly known as the thin-walled maze polypore, garnered attention due to its ecological role as a lignicolous fungus that contributes to the decay of injured hardwoods, especially willows and sapwood through white rot, where the lignin is degraded, leaving cellulose as a light-colored residue [23]. While this species is abundant across many Mediterranean countries, it is also frequently found in various forest ecosystems within Serbia [24]. This species has also garnered attention for its traditional uses, such as ornamental paper making, due to its unique textures and colors, showcasing its versatility beyond its ecological significance [24]. On the other hand, its pharmacological and medical importance has been overlooked aside from a few studies that have shown that this polypore fungus exhibits promising antibacterial, antifungal, and antioxidative activities [25,26,27].

Fungal bioactive compounds could possibly be increasingly used to help regulate hormonal balance, which is crucial for the reproductive health and overall well-being in women and thus could be beneficial during different stages of life, such as menstruation, pregnancy, or menopause. Furthermore, fungi are well-known for their immune-boosting properties, and a strong immune system is important for women’s reproductive health as it can help to fight off infections and maintain overall wellness, which is especially crucial during pregnancy [28,29,30]. The anti-inflammatory properties of mushrooms might help in reducing the inflammation in the body [31], potentially benefiting conditions like endometriosis or pelvic inflammatory disease. Adaptogenic fungi like reishi and cordyceps are believed to help the body adapt to stress. Managing stress levels is important for reproductive health as high stress can disrupt the menstrual cycle and fertility [32,33], while antioxidants play a role in overall health and may support reproductive health by protecting reproductive cells from oxidative stress [34]. Fungal species are touted for their energizing and revitalizing effects. Women dealing with fatigue or low energy levels due to hormonal changes or busy lifestyles may benefit from incorporating more fungi into their regular diet, probably due to polysaccharides, peptides, nucleosides, phenolic compounds, and triterpenoids, which mitigate human fatigue through effects on the functional systems, including the muscular, cardiovascular, hormone, and immune systems [35].

In this study, we highlight the diverse biomedical properties of the overlooked fungal species *D. confragosa*, emphasizing its antioxidant, cytotoxic, and enzyme modulation activities, among others, thus shedding light on its pharmacological significance for potential drug development and healthcare applications in the future. Importantly, we also propose its potential utilization in addressing women’s reproductive health, suggesting the use of oral formulations/teas or vaginalettes and hydrogels, leveraging its non-edible and woody characteristics. This emphasis on the gender dimension within the context of medicinal properties represents a novel aspect of our work in this field.

## 2. Results and Discussion

### 2.1. Extraction Yield

Following the extraction with 80% ethanolic (EtOH) and hydrodistillation-based EtOH extraction, the yields of the analyzed extracts from *D. confragosa* are summarized in Appendix A. The results revealed that the yield of the EtOH extract (DC) was significantly higher (0.31 ± 0.05 g/10 g d.w.) compared to that of the hydrodistillation-based EtOH extraction—DCHD (0.006 ± 0.05 g/10 g d.w.), indicating that the majority of the soluble components in the analyzed species were extracted using the EtOH maceration method. This contrasts with the findings of Rašeta et al. [36] for similar solvents but different autochthonous fungal species from Serbia (*T. versicolor* and *Stereum subtomentosum*), indicating the importance of the genotype regarding the overall bioactivity. The obtained yield of the DC extract was consistent with the EtOH and hot water extraction for *T. versicolor* and *S. subtomentosum* [36], while the DCHD extract had a lower yield in general. However, the anticipated low extraction yield was in line with expectations due to the characteristic of *D. confragosa* being a woody mushroom with minimal moisture content [37]. The results of Chandrawanshi et al. [38] depicted hot water as the best extracting solvent for *D. confragosa,* with a recovery percentage of 50.00, followed by EtOH (3.65%), while methanol (MeOH) provided only 2.70%. This is in accordance with the study where the highest yields for both basidiocarp/mycelial extracts of the same species were achieved using hot water extraction (21.9 ± 1.1% and 15.9 ± 0.7%, respectively). Conversely, the lowest yields were observed with MeOH (5.4 ± 0.5%) and 96% EtOH (6.2 ± 0.1%), respectively [26]. Since EtOH was employed as an inexpensive and proven solvent for extracting polar compounds with antioxidant potential [36], in this study, only this solvent was used.

### 2.2. Determination of Total Phenolic Content and Total Flavonoid Content Alongside Antioxidant Activity

The total phenolic content (TPC) and total flavonoid content (TFC) were determined in the DC and DCHD extracts of *D. confragosa*, alongside the DPPH radical scavenging capacity (RSC), ESR spectroscopy of the DPPH radical, and reduction capacity (Table 1). It is important to emphasize that hydrodistilled (HD) extracts of lignicolous fungi have not been extensively investigated chemically, nor has their biological activity been thoroughly evaluated [39]. Hence, to the best of our knowledge, this is the first report on the TPC, TFC, and antioxidant capacity of the *D. confragosa* HD extract.

The results indicate that the TPC and TFC were twice as high in the DC extracts (25.30 ± 1.05 mg GAE/g d.w. and 2.84 ± 0.85 mg QE/g d.w., respectively) compared to the DCHD extracts, while the TF/TP ratio was 11.22 and 8.33%, respectively. Phenolics were found to be more prevalent than flavonoids in both the fruiting bodies and mycelia extracts of *D. confragosa* from Serbia [26], which is in accordance with our results, where the TPC was higher in both the tested extracts (DC and DCHD). Compared to the TPC and TFC of *D. confragosa* from Croatia, the 50% EtOH extract contained 54.17 mg GAE/g of total phenols and 48.46 mg catechin/g of total flavonoids, while the TF/TP ratio was around 90% [37]. On the other hand, the MeOH extracts of *D. confragosa* from Poland contained lower phenolic content, with TPC = 6.88 ± 0.85 mg GAE/g d.w. [40]. The difference in these contents could be explained by the different extraction method, different assay obtained, as well as the higher polarity of the used solvent (i.e., 50% EtOH and MeOH) since higher extraction yields of phenolic compounds can be obtained by the increase of a solvent polarity [41]. Furthermore, according to Manjusha [42], the MeOH extract has been found to exhibit greater efficacy for the extraction of flavonoids compared to hot water and hexane extracts, while, for phenolics, the hot water extract is deemed superior. Conversely, among the EtOH, MeOH, and water extracts, the highest TPC was recorded in the 70% EtOH (126.40 ± 3.53 and 72.60 ± 3.82 μg GAE/mg d.w. in basidiocarp/mycelia, respectively), while the highest TFC were recorded in the 96% EtOH (1.21 ± 0.03 and 11.82 ± 0.47 μg QE/mg d.w.) [25]. The extraction rate was primarily influenced by the solvent used, with minimal dependence on the type of material used (fruiting body or mycelia) [26]. The results of this work also indicate that the extraction method plays a significant role in the extraction rate of the bioactive compounds, specifically related to hydrodistillation.

The DC extract demonstrated superior antioxidant activity compared to the DCHD extract, evidenced by its higher RSC values for the DPPH radical (IC_50_ = 8.53 ± 0.39 μg/mL) and IC_50_ = 0.981 mg/mL for the ESR spectroscopy. Moreover, the reduction capacity of the DC extract was found to be three times stronger than that of the DCHD extract (29.74 ± 0.81 and 9.24 ± 0.74 mg eq AAE/g d.w., respectively).

Compared to the analyzed extracts in this study, the scavenging capacity of the DPPH radical from the MeOH extract of *D. confragosa* from Bangladesh (IC_50_ = 51.21 μg/mL) was five times lower, attributed to the presence of phenolic compounds like flavonoids and tannins [35]. Conversely, the correlation between the DPPH RSC activity of the extracts from Serbia and the phenol/flavonoid contents was found to be very low [26], which is in contrast with our previous studies on other fungal species, such as *Ganoderma*, *T. versicolor,* and *S. subtomentosum* [31,36]. Moreover, phenolic acids (*p*-hydroxybenzoic acid, protocatechuic acid, and vanillic acid) are the dominant group of phenols and the carriers of antioxidative activity in *D. confragosa* and connected to a high TPC with reduction potential [40]. Hence, the enhanced antioxidant properties of the DC extracts can be attributed to the higher TPC and TFC detected since these compounds may serve a dual function in reducing the oxidation rate as they engage in both iron chelation and radical trapping processes [37,41,43]. On the other hand, the 50% EtOH extracts of *D. confragosa* from Croatia exhibited much lower neutralization of the DPPH radical (IC_50_ = 0.015 ± 0.07 mg/mL) [37], while the MeOH extracts of *D. confragosa* from India had even lower anti-DPPH activity of IC_50_ = 4.8 mg/mL [42]. Moreover, the EtOH extract of this species collected in India exhibited higher DPPH scavenging ability (IC_50_ of 18.07 ± 0.02 μg/mL), together with high redox capacity, compared to the hot water extracts, which gained higher yields [38], confirming the previously reported data [41] that the yield values are not directly related to their qualitative efficiency. Additionally, the variation in the antioxidant capacity, as well as the content of the bioactive compounds, may also be linked to factors such as the geographical origin, ecology, and genotype of the species [4,40]. Furthermore, the antioxidative potential of the basidiocarp extract of *D. confragosa* collected at Avala mountain in Serbia exhibited a direct correlation with its concentration, while, in the mycelia extracts, such a correlation was not evident since the scavenging activity of all the tested extracts declined at higher concentrations [26], indicating that the solvent together with the origin of the extract affect the antioxidative activity as well. 

In conclusion, the examined extracts of *D. confragosa* demonstrate antioxidant activity, a finding consistent with the previous research [25], albeit with variations noted across the different extraction techniques employed. In general, the choice of the extraction method together with the solvent used can significantly influence the antioxidant properties and bioactive compound content.

### 2.3. Cytotoxic Activity

The antiproliferative activity of the *D. confragosa* extracts on the estrogen-dependent human breast adenocarcinoma cell line—MCF-7 was investigated using standard in vitro MTT assay. Antiproliferative effect of both extracts in MCF-7 cells was concentration and time-dependent, reaching IC_50_ below 300 μg/mL (Table 1). The most potent activity was observed in the DCHD extract during the subacute incubation (after 24 h) and after the chronic incubation lasted for 72 h (IC_50_ values of 32.15 ± 0.99 μg/mL and 27.76 ± 0.89 μg/mL, respectively), while the DC extracts showed less inhibition of the MCF-7 cells (IC_50_ 88.23 ± 1.43 μg/mL and 81.45 ± 1.32 μg/mL, respectively). However, at 900 μg/mL, the DC/DCHD exhibited antiproliferative activity of 77.69% and 94.02%, respectively, suggesting that the compounds in the DCHD extract exert an almost 20% more potent cytotoxic effect. The results align with the existing literature regarding the cytotoxic effects of this and other fungal species [2,43,44,45]. One of the most active species in vitro against murine cancer cell line L1210, a lymphocytic leukemia, was the *D. confragosa* MeOH extract (IC_50_ = 74.5 μg/mL), while the in vivo antitumor activity was suggested to be linked to the presence of previously identified polysaccharides and beta-glucans [46]. On the other hand, the MeOH extracts of *D. confragosa* could serve as a potential source of cytotoxic substances due to the presence of various functional compounds like fatty acids, polyphenols, flavonoids, saponins, steroids, and alkaloids, among which certain fatty acids and polyphenols can disrupt the membrane structures, although the mechanism responsible for generating analgesic and cytotoxic effects is still under estimation [43]. The potential mechanism underlying the antiproliferative activity of the EtOH of *G. lucidum* on MCF-7 cells may involve apoptosis in human breast cancer cells, possibly mediated through the upregulation of the pro-apoptotic BAX protein pathway [44]. According to the previous data on the detected terpenoids in *G. applanatum* [47], aside from steroids, the primary components are probably responsible for the antiproliferative activity on MCF-7 cells, leading to apoptosis induction through various mechanisms, such as the inhibition of cell division or protein synthesis. Hence, we assume that diverse antiproliferative effects may arise from the combined actions of various compounds within the HD extract [2,45].

On the other hand, there is evidence that estrogen-dependent cancer cells, such as MCF-7, and cervical cancer cell lines, such as HeLa and SiHa, share common characteristics in terms of their response to estrogen and other hormones [48]. Moreover, the interplay of estrogen and progesterone, known for their impact on breast cancer cell proliferation [48], may extend to cervical cancer cells, given their expression of hormone receptors. This implies a connection between the estrogen-dependent cancer cells and cervical cancer cell lines, indicating their shared sensitivity to hormonal influences and potentially similar mechanisms governing growth and proliferation [48]. Therefore, exploring the effects of fungal extracts that contain well-known phytoestrogens such as genistein, daidzein, formononetin, glycitein, and biochanin [49], which have shown promise in modulating hormone-related pathways in breast cancer, may offer insights into their potential role in managing cervical cancer as well [50]. Having established the effectiveness of the *D. confragosa* extracts on estrogen-dependent breast cancer cells, future investigations should concentrate on evaluating their inhibitory impact on the cancer cell growth, promotion of apoptosis, and inhibition of proliferation in cervical cancer cell lines like HeLa and SiHa. Additionally, it is essential to conduct a detailed characterization of the recognized phytoestrogenic flavonoids as potential mycoestrogens from this underexplored medicinal fungal species. This is supported by the results presented (Table 1), indicating that *D. confragosa* contains flavonoids (1.01 to 2.84 mg QE/g d.w.). Overall, this preliminary study has shown that *D. confragosa* holds promise as a potential natural agent for combating women’s cancer through its anticancer activity.

### 2.4. Inhibition of AChE

The inhibition activity of AChE by two fungal extracts (DC and DCHD) was evaluated to assess their potential neuroprotective properties (Table 1). The AChE inhibition activity was expressed as IC_50_ relative to the control used (galantamine). The data revealed that both extracts exhibited significant AChE inhibitory effects compared to the control group (*p* < 0.05). However, there were notable differences in the inhibition potency between the two extraction methods.

The DCHD extract demonstrated a slightly higher AChE inhibition activity compared to the DC fungal extract. This difference could be attributed to variations in the composition and concentration of the bioactive compounds extracted through different methods [51]. The utilization of DCHD extraction could have enabled the extraction of a wider range of mycochemicals, such as polyphenols, flavonoids, and alkaloids, along with volatile unsaturated hydrocarbon compounds categorized as monoterpenoids, diterpenoids, sesquiterpenoids, and triterpenoids [52,53,54]. These compounds have been reported to exhibit AChE inhibitory properties [5,54]; hence, a possible synergistic effect could be responsible for the stronger inhibition of the AChE enzyme in the DCHD extract.

Furthermore, the mode of action of the AChE inhibitors present in the extracts could have influenced their inhibitory activity [55]. It is possible that the compounds present in the DCHD extract exert a stronger binding affinity towards the active site of AChE, resulting in slightly enhanced inhibition compared to those in the DC extract. Additionally, the differences in the solvent polarity and extraction conditions could have influenced the extraction efficiency and consequently the AChE inhibition activity. EtOH, being a polar solvent, may have facilitated the extraction of a wider range of bioactive compounds in the DCHD even though this extract had a lower yield. These findings corroborate an earlier study indicating that the yield values do not directly correlate with their qualitative effectiveness [41].

Overall, the results indicate that both fungal extracts possess AChE inhibitory activity, suggesting their potential utility in neuroprotective applications. It is important to emphasize that, to the best of our knowledge, this is the first report regarding the AChE inhibition of *D. confragosa*. Hence, further studies are warranted to elucidate the specific bioactive compounds responsible for the observed effects and to explore their therapeutic potential in neurodegenerative disorders.

### 2.5. Activity of PP1 Enzyme

The protein phosphatase-1 (PP1) activity of the DC and DCHD fungal extracts was evaluated to assess its potential as a modulator of cellular phosphoproteinase. Activity that was above 100% represented enhanced enzyme activity in the presence of the extract and was expressed as an activity factor above 1. The analyzed extracts showed the increased activity of the PP1 enzyme, i.e., the activity was greater than 100% (it had an activity factor above 1), where DC exhibited approximately three times stronger activation of the PP1 enzyme (Table 2). However, the results revealed intriguing insights into the potential differences in the bioactivity of the *D. confragosa* extracts extracted with different methods. The study highlighted that the simple EtOH extraction method yielded DC extracts with a broader spectrum of compounds, including those soluble in EtOH, which might influence the overall greater PP1 activity profile of the extract. On the other hand, the HD method, known for its selectivity in extracting terpenoids [51], could result in extracts enriched with specific bioactive components that could modulate the PP1 activity differently since lower activity was observed. This suggests that the distinct extraction procedures could influence the composition of the fungal extracts, affecting their PP1 inhibitory activity and overall bioavailability since it is known that the primary solvent characteristics, such as polarity and hydrophobicity, are often associated with enzyme activity and stability [56].

Moreover, the results indicate the possibility of using *D. confragosa* in tinctures that could stimulate the metabolism by the oral use of glycogen [57], so their action on PP1 could be one of the explanations for the mechanism of the antidiabetogenic action since it is known that this well-characterized Ser/Thr phosphatase plays a key role in regulating multiple cellular functions, such as glycogen metabolism [9].

The research results also suggest that the fungal extracts of *D. confragosa* may have potential as a component of a bio-preparation or dietary supplement for regulating blood sugar levels, particularly in pregnant women with gestational diabetes who do not yet require oral anti-diabetic therapy with insulin [12]. This is supported by studies that have shown that fungi have potential in anti-diabetes due to their ability to modulate the gut microbiota, ameliorate oxidative stress, beta-cell dysfunction, and insulin resistance, and inhibit the glucose absorption from diets, reduce hepatic glucose production, and increase the insulin sensitivity in tissues [58,59]. Moreover, incorporating fungi into the diet during pregnancy may offer benefits in managing GDM by supporting better glucose control and potentially reducing the risk of related complications. However, further research is needed to confirm the anti-diabetic effects of the *D. confragosa* extracts and to determine the optimal dosage and safety for human use.

### 2.6. Extracts’ Effect on DNA Integrity

The results of the extracts’ effect on DNA integrity are shown in Figure 1; Figure 2 and clearly indicate that the DCHD did not exhibit genotoxic activity, while a half-hour incubation of the DNA σ-2 virus of the DC extract could indicate DNA disintegration (Figure 1). However, the results and one-sided interpretation are hindered by the fluorescence that occurred due to the presence of UV-absorbing substances in the extracts themselves. Namely, it is likely that the strong fluorescence obtained in the work with basic concentrations masked the disintegration of the DNA. A preliminary test of the antiviral activity of the DC extract lasting half an hour indicated a short incubation time, so, in the repeated test, it was extended to 24 h against σ-2 and δ. The same trend, i.e., masking the disintegration σ-2 virus, was observed. On the other hand, the concentration of 1:100 was successful against the δ virus, considering that the disintegration of the DNA is clearly visible on the gel (Figure 2).

The successful disintegration of DNA integrity could be attributed to phenolic compounds that are characterized by the presence of an aromatic ring with hydroxyl substituents [60,61], whose presence is confirmed in the majority of mushrooms [4,5,6,36,45]. This could also explain the genotoxic activity of the DC extract, which showed a higher TPC compared to the DCHD extract, which did not show antiviral activity and had a lower TPC content. The ability of these biomolecules to act as antioxidants is attributed to their free hydroxyl group, which is attached to the phenolic ring. Based on the results presented in Table 1 and considering the antioxidative properties of the analyzed extracts, the DNA disintegration is probably not a consequence of free oxygen radicals but rather is a nonenzymatic degradation mediated by the direct interaction of the extract compounds with the DNA molecules. Rašeta et al. [36] in their study suggested that these results could indicate the potential genotoxicity of the extracts when they are applied at high concentrations or after prolonged exposure. Also, the authors concluded that numerous DNA protective and repair mechanisms are present in the eukaryotic cells; the observed properties of the extracts should be further examined in some cell-based systems [36].

However, phenolic acids are able to inhibit certain enzymatic systems as well, and, due to their lipid peroxidation ability [62], they are also used as anti-inflammatory agents [63]. Phenolic compounds, commonly found in plant-based foods and beverages, can damage DNA integrity by generating reactive oxygen species (ROS) and inhibiting DNA repair enzymes. This oxidative stress leads to DNA strand breaks, base modifications, and other genetic damage. Their interference with the DNA repair mechanisms exacerbates this damage, emphasizing the importance of understanding their impact for assessing the health effects and devising mitigation strategies [60,61].

### 2.7. Hemolytic Activity

The intensity of the hemolytic activity was graded and marked with a different number of “+” depending on the time it took for this activity to be registered by the drop in the erythrocyte absorption (Table 3 and Appendix A). The results obtained by the hemagglutination test are also shown in the same Table 3 as the result for hemolytic activity. It was found that the analyzed extracts did not show agglutination ability, i.e., binding of red blood cells, which could support the assumption that such extracts could be administered intravenously. The strongest hemolytic activity in a time of up to 5 min was demonstrated only for the DCHD extract. The time at which the 50% lysis of erythrocytes is reached is marked as the half-time of hemolysis t_50_, i.e., the time at which the absorption of the erythrocyte suspension obtained at 650 nm “falls” by 50%, while the concentration of this extract at that time is designated as HC_50_ (mg/mL) (Table 3). The lower concentration achieved indicates a stronger hemolytic activity.

The hemolytic activity was assessed using the control solvent (EtOH); however, no activity was detected, suggesting that the observed hemolytic activity in the crude extract is likely attributable to the active components derived directly from the extracts themselves (Table 3). The activity was significantly affected by the increased concentration of the extract itself, as well as the type of solvent, but also the method of extract preparation (e.g., simple maceration with the extract obtained after the HD procedure), which can be observed by comparing the different extracts. Thus, increased activity after HD was observed compared to the DC extract. This could indicate the diversity and specificity of the active compounds present in this fungal species, and especially the increase in the amount of polar glycosides in the HD extracts, which could be responsible for the achieved effects.

Since the organic extract that showed higher hemolytic activity was prepared in ethanol solvent after the HD procedure, the presence of terpenoid compounds or terpenoid or steroid aglycones (sapogenins) can be expected in such extracts, which have already been identified in *Daedaleopsis* sp. petroleum and MeOH extracts [42,64]. Even though Fakoya et al. [64] did not detect tannins and phenols in the EtOH extract of *D. confragosa* from Nigeria, this solvent is effective for the extraction of terpenoids, especially after the HD procedure. Hence, we assume that the DCHD extract also likely contains saponins, which are glycosides containing terpenoids or steroids [45,65]. This assumption was based on previous research, where the reactions typical of the detection of saponins themselves, observed in the species *Meripilus giganteus*, were foaming along with hemolysis of the cell membrane of erythrocytes [66]. The hemolytic reaction of erythrocytes is believed to occur due to the affinity of the aglycone (terpenoid or steroid) to membrane sterols, particularly cholesterol, forming insoluble complexes. The hemolytic activity of saponins tends to increase with the number of polar groups in the aglycone, with single-chain steroidal and triterpenoid saponins displaying greater activity compared to those with two chains. Hemolysis also involves the formation of a saponin–cholesterol complex, alterations in the organization of the phospholipids in the sarcolemma membrane, and the generation of products such as phosphatidic acid, all dependent on the structural configuration and three-dimensional orientation of the saponins [67]. Hence, further analysis of these compounds in various *D. confragosa* extracts should be conducted.

The results of the research related to hemolytic activity suggest that potential preparations based on the analyzed fungal extracts may be in the form of oral or vaginal formulations along with intravenous administration. Rafique et al. [68] suggest that oral and vaginal formulations are safer than parenteral ones due to the lower risk of hemolytic reactions despite Pino et al. ‘s [69] findings regarding the oral or vaginal administration of a plant extract being non-hemolytic, whereas the intravenous administration was hemolytic. Furthermore, the anti-hemolytic effects of the fungal extract may operate through distinct mechanisms compared to those of plant extracts. Our findings indicate that all three modes of administration might be considered safe as they do not demonstrate agglutination capability (Table 3), but further confirmation is needed in the future.

## 3. Materials and Methods

### 3.1. Fungal Material

*D. confragosa* (Class Basidiomycetes, Order *Polyporales*, and Family *Polyporaceae*) was gathered in 2022 in Papratski do (Fruška Gora Mountain) in Serbia. The identification of these fungal species involved a comprehensive examination of their macroscopic features, as well as microscopic analysis. The voucher specimen was stored in the Fungarium under voucher No12-01053 at the ProFungi Laboratory (Department of Biology and Ecology, Faculty of Sciences, University of Novi Sad). One portion of the samples was frozen (−80 °C) and saved for further analysis.

### 3.2. Preparation of Fungal Extracts

Moreover, 80% EtOH fungal extract (DC) was prepared as follows: 10 g of dried and ground cleaned fruiting bodies were weighed to which 350 mL of 80% EtOH was added (Zorka, Šabac, Serbia). Maceration was carried out for 24 h, in the dark at room temperature. After that, the extracts were filtered through cheesecloth and filter paper. Further lyophilization was applied at −40 °C to −50 °C (Christ Alpha 1-2 LD Freeze Dryer, Bremgarten, Switzerland).

The process of hydrodistillation (HD) of fresh fungal material was conducted as follows: approx. 250 g of fungal material was poured with the appropriate amount of water (H_2_O) (approximately about 500 mL), and the process itself lasted 4 h at room temperature from the moment of boiling. Nonpolar compounds such as terpenoids were probably collected in hexane (2 mL) and stored in a refrigerator. After the hydrodistillation was completed, the aqueous layer was removed, and the material was transferred to an Erlenmeyer flask and poured with 1.5–2 times the volume of absolute ethanol (EtOH), which was later evaporated (Bűchi R-210, Flawil, Switzerland) to dryness and redissolved in 80% EtOH.

The dried residues of both extract types were dissolved in 80% EtOH to achieve a concentration of 100 mg/mL and then stored at +4 °C for future utilization.

All fungal extracts were prepared in ProFungi Laboratory at the Department of Biology and Ecology and in the Biochemistry Laboratory at the Department of Chemistry, Biochemistry and Environmental Protection at Faculty of Sciences, University of Novi Sad.

### 3.3. Determination of TPC and TFC and Antioxidant Capacity

The TPCwas measured using the Folin–Ciocalteu (FC) technique outlined by Singleton et al. [70], which relies on spectrophotometric detection of phenols forming a colored complex with an FC reagent. Absorbance was assessed at 760 nm (Multiskan Ascent, Thermo Electron Corporation, Waltham, MA, USA), and TPC was expressed as milligrams of gallic acid equivalents (GAE) per gram of dry weight (d.w.) (mg GAE/g d.w.) using a standard calibration curve.

Similarly, the TFC) was determined spectrophotometrically using aluminum chloride (AlCl_3_) as the complexing agent for flavonoids. Absorbance readings were recorded at 415 nm, and quercetin served as the standard for calibration. The TFC value of the extract is reported as milligrams of quercetin equivalents (QE) per gram of d.w. (mg QE/g d.w.) [71].

The RSC of fungal extracts on DPPH^•^ was evaluated in vitro using the method outlined by Espin et al. [72] and by Electron Spin Resonance (ESR) spectroscopy.

DPPH scavenging activity was evaluated following the procedure detailed by Espin et al. [72], which involves the conversion of purple DPPH^•^ (2,2-diphenyl-1-picrylhydrazyl radical) into its reduced yellow form, DPPH-H, after a 30 min incubation at room temperature without exposure to light. Absorbance was then measured at 515 nm. The results are expressed as IC_50_ values, which represent the concentration at which 50% inhibition of the free radicals is achieved.

The ESR antioxidant assay is utilized to evaluate the antioxidant activity of extracts, employing the reduction of the DPPH radical as outlined in Karaman et al. [41]. To determine the antioxidant activity, a solution of stable DPPH free radicals was prepared by mixing 200 μL of H_2_O and 400 μL of a 0.4 mM MeOH solution of DPPH (blank test) (Bruker 300E ESR spectrometer, Bruker, Billerica, MA, USA). The influence of fungal extract on the transformation of DPPH radicals was analyzed in a solution obtained by mixing a certain concentration of extracts that were in the range of 0.25–12.5 mg/mL in the final volume. The mixture was vigorously stirred for 2 min and transferred to a Bruker ER-160FC quartz cuvette for aqueous solutions and ESR spectra were recorded under the following conditions: modulation frequency 100 kHz, modulation amplitude 0.226 G, time constant 40.96 ms, measurement time range 671.089 ms, field center 3440.00 G, total measuring range 100.00 G, microwave frequency 9.64 GHz, current strength 5.00 × 10^5^, microwave power 20 mW, and measuring temperature 23 °C.

RSC was defined using Equation (1) as follows:
RSC_DPPH_ = (h_0_ − h_x_) × 100/h_0_ (%)(1)
wherein
h_0_—the height of the second peak of the DPPH radical ESR signal of the blank sample;h_x_—the height of the second peak of the ESR signal of the DPPH radical of the sample that contains extract.

Redox capacity of the analyzed extracts was completed using the ferric reducing antioxidant power (FRAP) assay, which includes the reduction of FeIII–TPTZ (Iron (II)-2,4,5-tripyridyl-S-triazine) under acidic conditions, leading to the creation of a blue FeII–TPTZ complex. This process is measured by assessing the absorbance of the resultant intense blue complex at 593 nm, following the method described by Benzie and Strain [73]. The results were quantified and expressed as mg ascorbic acid equivalents per gram of d.w. (mg AAE/g d.w.).

### 3.4. Cytotoxic Activity

The cytotoxic effectiveness of the examined fungal extracts was evaluated utilizing the estrogen-dependent breast cancer cell line (MCF-7), following the protocol described by Mosmann [74]. Cancer cell viability was tracked over both 24 h (acute) and a 48 h (sub-acute) incubation periods, encompassing extract concentrations ranging from 50 to 250 µg/mL. Cell cytotoxicity was determined by the IC_50_, representing the concentration that inhibits 50% of cell growth, extrapolated from concentration–response curves.

### 3.5. Enzyme Assays

#### 3.5.1. Anti-Acetylcholinesterase Activity

Inhibition of AChE enzyme was determined according to Ellman et al. [75] assay adapted to the use of microtiter plates with 96 wells, as described by Mišković et al. [4]. For the degradation of the AChE enzyme, acetylcholine iodide (AChI) was employed as a substrate. The reaction mixture consisted of 20 µL of the fungal extract, 150 µL of reagent A (comprising Ellman’s reagent and AChI), and 50 µL of reagent B (AChE enzyme at 518 U/mL previously dissolved in phosphate buffer, pH 8.0), while donepezil at 1 mg/mL (Donecept, Zdravlje, Leskovac, Serbia) served as a positive control. Absorbance was measured at 412 nm using a 96-well plate reader (Multiskan Ascent, Thermo Electron Corporation, USA) over a duration of 15 measurements with 1 min intervals. The percentage (%) of enzyme inhibition was determined using Equation (2) as follows:
I_AChE_ (%) = (1 − A_sample_/A_control_) × 100% (2)
where A_sample_ and A_control_ stand for the absorbance of the tested and control samples, respectively. All measurements were conducted in triplicate, and the results were expressed as IC_50_ values, with lower values indicating higher AChE activity of the sample. The reference inhibition time was set at 10 min or 600 s, respectively.

#### 3.5.2. Activity of Protein Phosphatase-1 Enzyme

The activity of PP1 and its inhibition by the analyzed samples were determined by the colorimetric method following the method described by An and Carmichael [76]. Recombinant PP1 enzyme expressed in *Escherichia coli* was employed in the assay. The extracts were tested at final concentrations of 5, 50, 250, and 500 μg/mL. Further, 10 μL of PP1 enzyme and 10 μL of extract were added to each well of a 96-well plate and incubated for 5 min at room temperature. The enzyme was previously diluted in a buffer containing 50 mM Tris-HCl (pH 7.4), 1 mM MnCl_2_·4H_2_O 2-mM DTT (dithiothreitol), and 1 mg/mL BSA (bovine serum albumin). Subsequently, 180 μL of *p*-nitrophenyl phosphate (pNPP) substrate diluted in a buffer containing 50 mM Tris-HCl (pH 8.1), 20 mM MgCl_2_·6H_2_O, 0.2-mM MnCl_2_·4H_2_O, and 0.5 mg/mL BSA was added to each well. Deionized H_2_O served as the negative control, while the pure toxin microcystin MC-LR was utilized as the positive control. After a 2 h incubation period at 37 °C, PP1 activity was determined by measuring the absorbance of the yellow product *p*-nitrophenol (pNP) at 405 nm using a microplate reader.

### 3.6. Effect of Fungal Extracts on Viral DNA Integrity

This assay was conducted according to method described in Rašeta et al. [36], whereas DNA was extracted from *Pseudomonas aeruginosa* phages σ-2 (family Siphoviridae) and δ (family Podoviridae) [77], previously purified via CsCl gradient, was subjected to the phenol-chloroform extraction method [78]. The impact of undiluted (1:1), 1:100, and 1:10,000 diluted fungal extracts on the stability of viral DNA was evaluated. These mixtures were incubated for either 30 min or 24 h at 37 °C, and the integrity of the DNA was assessed by analyzing agarose gel electrophoresis, with results documented accordingly.

### 3.7. Hemolytic Activity

The assay was performed according to the method optimized by Sæbø et al. [79], where erythrocytes used in this assay were isolated by centrifugation from fresh bovine blood to which citrate was added during delivery, in order to avoid blinding. Erythrocytes were washed three times with physiological solution and used for biological tests or placed in Alsever’s preservative in the refrigerator. The erythrocytes prepared in this way can be used until the supernatant turns red, indicating that hemolysis has occurred. Before use, preserved erythrocytes were washed twice with saline. For testing, they were resuspended in an erythrocyte buffer (a solution of 0.13 M NaCl in 0.02 M TRIS HCl), pH 7.4. The absorption of the prepared erythrocyte suspension at 650 nm was 0.5 ± 0.01. Hemolytic activity was determined with the help of a microtiter plate reader, which enables simultaneous reading of 96 samples. Further, 96 microtiter plate wells were filled with 100 μL of erythrocyte buffer and 20 μL of sample (fresh or boiled extract). After pipetting, another 100 μL of erythrocytes were added to each well and quickly measured. Hemolysis was observed as a decrease in absorbance at 650 nm. For control, 100 μL of erythrocyte buffer, 20 μL of deionized water, and 100 μL of erythrocytes were used. Hemolysis was monitored for 20 min at 25 °C. In the case of samples that were active, dilutions were determined, and half the hemolysis time (t_50_) was read, i.e., the time at which absorption falls to half of its initial value. Dilutions were conducted in proportions 1:1, 1:2, 1:10, 1:20, 1:50, 1:80, and 1:100.

### 3.8. Statistical Analysis

All experiments were conducted in triplicate and reported as mean ± standard error (*n* = 3), except hemolytic assay. Statistical analysis to determine differences among groups was performed using a one-way analysis of variance (ANOVA, with post hoc Tukey’s test) test with SPSS program (version 16.00). Significance levels were set at *p* < 0.05.

## 4. Conclusions

In summary, as far as our understanding extends, this study represents the first report of *D. confragosa*’s anti-AChE activity, likely attributed to its steroid and terpenoid compounds, or their aglycons within saponins, which also demonstrate high hemolytic and cytotoxic activity. Conversely, concerning the identified antioxidant activity, as well as the activity of PP1 and genotoxic activity, it is likely that other types of secondary metabolites, particularly phenolics, predominantly contribute to these effects. Also, this study proves that the bioactivity of *D. confragosa* extracts is heavily influenced by the extraction methods, while positive bioactivity profiles, including antioxidant, cytotoxic, and genotoxic activities, and enzyme modulation, were observed. Hence, this study underscores the potential for the development of therapeutic interventions or supplements aimed at enhancing women’s reproductive health utilizing *D. confragosa* or its bioactive compounds. However, further exploration of *D. confragosa* and its bioactive compounds for the development of therapeutic interventions or supplements targeting women’s reproductive health are needed, along with continued research, including clinical studies, to validate and translate the observed bioactivity profiles into practical applications for women’s health.

## Figures and Tables

**Figure 1 pharmaceuticals-17-00600-f001:**
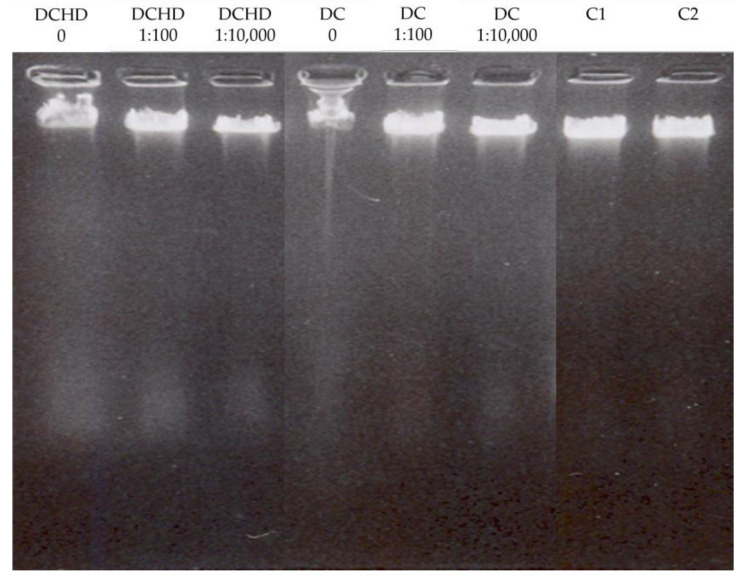
Electrophoretic separation of genomic DNA of σ-2 virus incubated 30 min with fungal extracts (DC—*D. confragosa* EtOH extract; DCHD—*D. confragosa* hydrodistillation-based EtOH extract). Concentration of extracts: 0, 1:100, and 1:10,000. Control: C1—viral DNA and DMSO; C2—viral DNA.

**Figure 2 pharmaceuticals-17-00600-f002:**
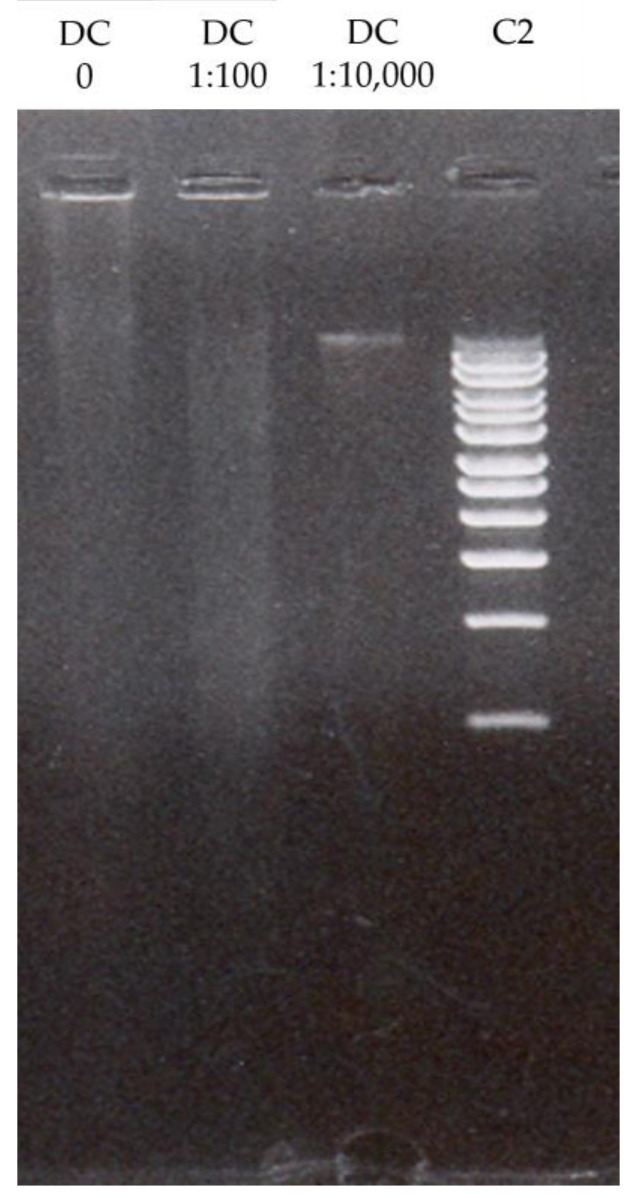
Electrophoretic separation of genomic DNA of δ virus incubated one day with *D. confragosa* DCHD extract. Concentration: 0, 1:100, and 1:10,000. C2—untreated viral DNA with DMSO.

**Table 1 pharmaceuticals-17-00600-t001:** Analysis of *D. confragosa* EtOH extracts for total content (TPC and TFC), antioxidant capacity, anti-acetylcholinesterase activity, and cytotoxic activity.

	Assay	*D. confragosa*	
	EtOH–DC	HD–DCHD	
**Total Content**			
TPC (mg GAE/g d.w.)	25.30 ± 1.05	12.12 ± 0.89	
TFC (mg QE/g d.w.)	2.84 ± 0.85	1.01 ± 0.13	
**Antioxidant Activity**			
DPPH (IC_50_ (μg/mL))	8.53 ± 0.39	10.13 ± 0.76	
ESR DPPH (IC_50_ (μg/mL))	0.981	1.565	
FRAP (mg AAE/g d.w.)	29.74 ± 0.81	9.24 ± 0.74	
**Enzyme Modulation**			**Galantamine** *
Anti-AChE (IC_50_ (μg/mL))	3.59 ± 0.78	3.11 ± 0.45	0.08 ± 0.002
**Cytotoxic Activity**			**Ellagic Acid** **
MTT (IC_50_ (μg/mL)) 24 h	88.23 ± 1.43	12.15 ± 0.99	35.82 ± 1.52
MTT (IC_50_ (μg/mL)) 72 h	81.45 ± 1.32	27.76 ± 0.89	40.07 ± 0.97

TPC—total phenolic content; TFC—total flavonoid content; HD—extract made by hydrodistillation technique; * galantamine was used as a standard compound; MTT assay was conducted on MCF-7 cell lines; ** ellagic acid was used as a standard compound. The data are presented as the mean ± s.d. of triplicate measurements.

**Table 2 pharmaceuticals-17-00600-t002:** The inhibition of protein phosphatase 1 (PP1) by analyzed *D. confragosa* EtOH extracts.

Parameters	*D. confragosa*		Control
	EtOH–DC	HD–DCHD	80% EtOH
Basic concentration (mg/mL)	125	125	-
Final concentration (mg/mL)	3.0	3.0	-
AE PP1 (%)	339.39 ± 1.16	217.42 ± 1.09	46.96 ± 0.65
AF	3.39	2.17	0.47

HD—this extract is prepared using the hydrodistillation technique; AE—enzyme activity; AF—factor of activity.

**Table 3 pharmaceuticals-17-00600-t003:** Examination of hemolytic activity of analyzed *D. confragosa* EtOH extracts.

Parameters	*D. confragosa*		Control
	EtOH–DC	HD–DCHD	80% EtOH
Tested concentration (mg/mL)	125	125	-
t_50_ (min)	<50	>20	0
Intensity of hemolysis	+, −, colored	++++	-
Hemagglutination (HGA)	-	-	-
Tested concentration (mg/mL)	n.a.	150	-
t_50_ (min)	n.a.	6.06	n.a.
1/t_50_ (min^−1^)	n.a.	0.17	n.a.
HC_50_ (mg/mL)	n.a.	81.0	n.a.

HD—this extract is prepared using the hydrodistillation technique; t_50_—represents the time of half-hemolysis; 80% EtOH—control; “+,−”—hemolysis started after 20 min; “++++”—hemolysis took place within 5 min; n.a.—not analyzed.

## Data Availability

The data presented in this study are available as Appendix A.

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
