# Peer review of "Bioactivity Profiling of Daedaleopsis confragosa (Bolton) J. Schröt. 1888: Implications for Its Possible Application in Enhancing Women’s Reproductive Health"

_pharmaceuticals, 2024, doi:10.3390/ph17050600_

Round 1

Reviewer 1 Report

Comments and Suggestions for Authors

Dear Authors,

Review of the paper titled: "Bioactivity Profiling of Daedaleopsis confragosa (Bolton) J. 3 Schröt. 1888: Implications for Its Possible Application in 4 Enhancing Women's Reproductive Health”

Detailed Remarks:

1.   The abstract is well-structured.

2.   The introduction is too lengthy and should end clearly with the presentation of the study's aim or an alternative research hypothesis in contrast to the null hypothesis, followed by its verification in the subsequent sections.

3. The results of the study have been discussed in the context of potential mechanisms of action of bioactive compounds present in mushroom extracts. The authors identified various chemical components, such as terpenoids and steroids, which may be responsible for the observed biological properties. Additionally, possible implications of the results for women's reproductive health were also presented, such as the potential antioxidative, cytotoxic, and genotoxic effects of mushroom extracts. It is worth noting that the interpretation of the results also considered the limitations of the study, such as the lack of evaluation of individual chemical components in assessing the bioactivity of specific extracts. This approach helps readers understand the significance and context of the obtained results and their implications for further research or potential practical applications.

4.  The "Materials and Methods" section should typically follow the introduction, which is the case in this manuscript. The authors have provided detailed and well-explained methodologies, including information on mushroom sourcing, extraction procedures, bioactivity analysis, and experimental conductance. Each step of the process is meticulously described, incorporating references to standard research methods and analytical techniques, which enhances the credibility of the experiments conducted. However, certain areas could benefit from further clarification or supplementation:

- Extraction protocol: While the ethanol extraction and hydrodistillation processes are detailed, specific extraction conditions such as time, temperature, solvent-to-material ratio, and final extract quantity are lacking. Providing additional information on these aspects would facilitate experiment replication by other researchers.

- Quality control: Although bioactivity analysis methods are generally described, there is insufficient information regarding sample quality control, including contamination checks, purity, and extract stability. Implementing robust quality control measures is essential for ensuring result reliability.

-   Statistical procedures: While the authors mention conducting experiments in triplicate and present results as mean ± standard error, details on the statistical procedures employed, such as the choice of tests or inter group comparative analysis, are missing. Including more information on this aspect would aid in result interpretation and reliability assessment.

5.  The discussion of the results is appropriately conducted, with thorough analysis considering research objectives, practical applications, and implications within the existing literature. Both strengths and limitations of the research are addressed, along with suggestions for further investigation or analysis, demonstrating a critical approach. Additionally, proposed mechanisms of action for bioactive compounds and their potential health benefits provide valuable context and understanding for readers.

6.  The conclusions drawn encompass both positive aspects, such as antioxidative properties, and potential risks, such as cytotoxic effects, highlighting the need for further research into bioactive compound mechanisms and therapeutic applications. These conclusions are pertinent for advancing research in this field and may inspire further scientific exploration into mushroom-derived compounds for women's reproductive health improvement. Conclusions should be a separate chapter in the work.

7. References cited are correctly selected and cover a broad spectrum of topics related to mushroom pharmaceutical properties and natural product research, providing valuable insights into their pharmacological potential across various therapeutic areas, including cancer treatment, diabetes management, and antimicrobial activity, along with elucidating their mechanisms of action.

Comments on the Quality of English Language

 Minor editing of English language required.

Author Response

 Dear  Reviewer, 

 Sincerely yours

authors

Reviewer 2 Report

Comments and Suggestions for Authors

The manuscript "Bioactivity Profiling of Daedaleopsis confragosa (Bolton) J. Schröt. 1888: Implications for Its Possible Application in Enhancing Women's Reproductive Health" is devoted to prediction an investigation of ethanolic and hydrodistillation based extracts of wood-rotting fungal species from Serbia. Free radical scavenging capacity, antioxidant activity with FRAP, total phenolic content, and total flavonoid content were determined. Such bioactivity tests as hemolytic, cytotoxic, and inhibition of the AChE enzyme experiments etc. were also performed.

This research is quite interesting for the specialists in biotechnology and pharmacognosy.

The manuscript is written in a good language, has a good structure and a clear representation of the data. I think, this manuscript can be published in the Pharmaceuticals journal after minor revision after taking into account the comments given below:

1.      Hemolytic activity method looks a little bit strange. Concentration dependence of hemolysis should be presented. Usually, absorbance of supernatant is measured. Please, provide the references for this method.

2.      Conclusion section is needed.

3.      “Enhancing Women's Reproductive Health” – this does not follow clearly from the materials of the work.

Author Response

 Dear  Reviewer, 

 Sincerely yours

 authors

Reviewer 3 Report

Comments and Suggestions for Authors

Please see attached a few recommendations for the authors

Comments on the Quality of English Language

Only minor quality check is needed

Author Response

 Dear  Reviewer, 

 Sincerely yours

 Authors
